# Level and Determinants of Adherence to COVID-19 Preventive Measures in the First Stage of the Outbreak in Uganda

**DOI:** 10.3390/ijerph17238810

**Published:** 2020-11-27

**Authors:** Bob O. Amodan, Lilian Bulage, Elizabeth Katana, Alex R. Ario, Joseph N. Siewe Fodjo, Robert Colebunders, Rhoda K. Wanyenze

**Affiliations:** 1Uganda Public Health Fellowship Program, Kampala 7272, Uganda; lbulage@musph.ac.ug (L.B.); ekatana@musph.ac.ug (E.K.); riolexus@musph.ac.ug (A.R.A.); 2Global Health Institute, University of Antwerp, Doornstraat 331, 2610 Antwerp, Belgium; JosephNelson.SieweFodjo@uantwerpen.be (J.N.S.F.); Robert.colebunders@uantwerpen.be (R.C.); 3School of Public Health, College of Health Sciences, Makerere University, Mulago Kampala 7072, Uganda; rwanyenze@musph.ac.ug

**Keywords:** COVID-19, preventive measures, adherence, satisfaction, Uganda

## Abstract

We conducted an online survey in the first two months of the Coronavirus Disease 2019 (COVID-19) epidemic in Uganda to assess the level and determinants of adherence to and satisfaction with the COVID-19 preventive measures recommended by the government. We generated Likert scales for adherence and satisfaction outcome variables and measured them with four preventive measures, including handwashing, wearing face masks, physical distancing, and coughing/sneezing hygiene. Of 1726 respondents (mean age: 36 years; range: 12–72), 59% were males, 495 (29%) were adherent to, and 545 (32%) were extremely satisfied with all four preventive measures. Adherence to all four measures was associated with living in Kampala City Centre (AOR: 1.7, 95% CI: 1.1–2.6) and receiving COVID-19 information from health workers (AOR: 1.2, 95% CI: 1.01–1.5) or village leaders (AOR: 1.4, 95% CI: 1.02–1.9). Persons who lived with younger siblings had reduced odds of adherence to all four measures (AOR: 0.75, 95% CI: 0.61–0.93). Extreme satisfaction with all four measures was associated with being female (AOR: 1.3, 95% CI: 1.1–1.6) and health worker (AOR: 1.2, 95% CI: 1.0–1.5). Experiencing violence at home (AOR: 0.25, 95% CI: 0.09–0.67) was associated with lower satisfaction. Following reported poor adherence and satisfaction with preventive measures, behavior change programs using health workers should be expanded throughout, with emphasis on men.

## 1. Introduction

The Coronavirus Disease 2019 (COVID-19) caused by the Severe Acute Respiratory Syndrome Coronavirus-2 (SARS-CoV-2) was declared a pandemic by the World Health Organization (WHO) in March 2020 [1]. Countries were urged to institute preventive strategies to minimize viral transmission. Accordingly, the Ugandan government progressively implemented several stringent public health measures to prevent and contain any local COVID-19 epidemic. By 18 March 2020, the Ugandan President banned all public gatherings and encouraged the public to observe physical distance, not to cough, sneeze or spit in public, and to observe strict hygienic rules (handwashing with soap and water or using sanitizers, regularly disinfecting surfaces, such as tables and door handles among others) [2]. The country further banned travel to and from other countries that had a large number of COVID-19 cases. Not only did the President suspend discos, bars, sports, cinemas, dances, and concerts, but also discouraged extravagant weddings that attracted large numbers of people. Later, on 20 March 2020, all institutions of learning were closed. The first case of COVID-19 in Uganda was reported on 21 March 2020. On 25 March 2020, a ban on public transport was instituted, and all passenger movements into Uganda by air, land, or water were stopped following reports of multiple escapes of people from mandatory quarantine centers. A 14-day total lockdown started on 30 March 2020, with a nationwide curfew from 7 p.m. to 6.30 a.m.; the use of private cars was equally banned, except for essential staff. The lockdown was later extended until 2 June 2020, when a phased easing of the restrictions commenced.

However, the adherence to personal preventive measures, such as physical distancing, mask use, hand and cough hygiene, were not evaluated. Understanding the level of adherence to and satisfaction with personal preventive measures is essential for the containment of the COVID-19 epidemic in the long-term. We assessed the level and determinants of adherence as well as the population’s satisfaction with respect to the COVID-19 preventive measures recommended by the government.

## 2. Materials and Methods

### 2.1. Study Design and Population

We conducted a cross-sectional national survey as part of the International Citizen Project (ICP) to assess adherence to preventive measures and their impact on the COVID-19 outbreak. The ICP consortium created a generic questionnaire to investigate the impact of COVID-19 and associated restrictions on populations living in low and middle-income countries [3]. We modified this questionnaire based on the local situation in Uganda by adding questions that probed for the practice of disinfecting phones, bag handles, laptops, door keys/locks, and TV remotes. Additionally, the questionnaire was modified to probe for the level of satisfaction with each personal COVID-19 preventive measure. Other questions added were those that probed for reasons for discontinued medication when one had an underlying disease. The questionnaire collected information about socio-demographic characteristics; the impact of COVID-19 and associated restrictions on daily life, professional life, and personal well-being; adherence to personal and community preventive measures, and acceptability of these measures. The questionnaire (in English) was hosted securely on the study website (https://www.icpcovid.com), and the web-link was widely shared during the lockdown period via emails and social media platforms, such as WhatsApp, Facebook, and Twitter from 16 to 30 April 2020. The emailing list of District health officers was obtained from the Ministry of Health. District health officers from all over the country were given the survey web-link by email, and thereafter, they were asked to share with all district public servants and any other person outside local government. Whoever got the link was asked to disseminate it further, share it with any other persons in their networks. Moreover, the Facebook and Twitter platforms of the Ministry of Health and Makerere University School of Public Health actively shared the survey web-link. People with access to the internet either on smartphones or computers were able to voluntarily participate in the study by clicking on the link and anonymously submitting their responses.

### 2.2. Study Design, Study Variables, Data Management, and Data Analysis

To determine the overall level and determinants of adherence to the preventive measures, we generated a composite outcome variable called “overall level of adherence” using a 4-item Likert scale (1 = very poor adherence to 4 = high adherence). We generated this composite adherence outcome using the following four variables, each having a weight of 1: frequent handwashing (Many times in a day after contact with persons or surfaces), wearing face masks, physical distancing, and covering mouth or nose with tissue paper or fabric when coughing/sneezing. These four variables were selected since they were considered the most effective COVID-19 prevention measures [4]. Very poor adherence, represented by score 1 of the Likert scale, meant that the person did not adhere to more than one of the four preventive measures. Poor adherence, score 2 of the Likert scale meant that the person adhered to two out of the four major preventive measures. Moderate adherence, score 3 of the Likert scale meant that the person adhered to three out of the four major preventive measures. Adherence to all the four preventive measures was categorized as high adherence.

To determine the overall level and determinants of satisfaction with the preventive measures, we generated a 5-item Likert scale composite outcome variable of the overall level of satisfaction (1 = very dissatisfied to 5 = very satisfied) with the four preventive measures. Very dissatisfied, which is represented by score 1 on the Likert scale, meant that the person was extremely satisfied with less than one of the four preventive measures. Dissatisfied, score 2 on the Likert scale meant that the person was extremely satisfied with only one of the four preventive measures. Neutral, score 3 on the Likert scale meant that the person was extremely satisfied with only two of the four preventive measures. Satisfied, score 4 on the Likert scale meant that the person was extremely satisfied with only three of the four preventive measures. Extremely satisfied with all the four preventive measures was categorized as very satisfied.

Independent variables, including socio-demographic, daily personal health, and professional factors, were included as independent determinants of adherence and level of satisfaction to preventive measures. Using principal component analysis, we generated a composite variable on wealth index quintiles from household-item possession variables, such as possession of a car, television set, radio, etc.

Descriptive statistics were generated using means with standard deviation (SD) for continuous outcomes and percentages (%) for categorical variables. We summarized the number of times of handwashing in a day and the extent of adoption of the preventive measures using mean and SD.

We used ordinal logistic regression to determine the factors associated with adherence and satisfaction with preventive measures. We considered a *p*-value of <0.05 to determine the level of significance and a stepwise approach to ascertain the best fitting model. During multivariate analysis on the level of adherence to preventive measures, variables, including working from home, and flu-like symptoms, were dropped because of collinearity.

Data extracted from the secure server were cleaned using Microsoft Excel version 2013 and thereafter analyzed using Stata/SE 14.

### 2.3. Ethical Considerations

The ICP study was approved by the ethics committees of the University of Antwerp, Belgium (20/13/148), and School of Public health, Makerere University, Kampala, Uganda (HDREC number 809). All participants consented and entered their data anonymously.

## 3. Results

### 3.1. Participants’ Characteristics

A total of 1726 persons participated in the study, mean age of 36 years (range = 12 to 72). The majority of the respondents (59%) were males; only 47 (3%) of participants were non-Ugandans (Table 1).

### 3.2. Level of Adherence to the COVID-19 Preventive Measures in the First Stage of the Outbreak, Uganda

Only 495 (29%) of participants were adherent to all the preventive measures. However, there was a high level of adherence to some of the individual preventive measures. Overall, 96% adhered to frequent handwashing, 90% to physical distancing, and 86% to cough hygiene, whereas the use of masks was low at 33%. Other preventive measures with low adherence included disinfecting phone (42%), Laptop (26%), bag (20%), and TV remote (18%) (Table 2).

### 3.3. Determinants of Adherence to the COVID-19 Preventive Measures in the First Stage of the Outbreak, Uganda

In multivariable analysis, participants living in the Kampala City Centre (AOR: 1.7, 95% CI: 1.1–2.6), those who obtained COVID-19 information from healthcare workers (AOR: 1.2, 95% CI: 1.01–1.5), those who obtained COVID-19 information from village leaders (AOR: 1.4, 95% CI: 1.02–1.9), or those worried about their health (AOR: 1.5, 95% CI: 1.1–1.9) were more likely to adhere to the preventive measures positively. Staying with siblings reduced the odds for high adherence (AOR: 0.75, 95% CI: 0.61–0.93) (Table 3).

### 3.4. Level of Satisfaction with the COVID-19 Preventive Health Measures in the First Stage of the Outbreak, Uganda

Overall, 545/1726 (32%) of the participants were very satisfied with the preventive measures. Most [1251/1726 (73%)] of the participants were extremely satisfied with the measure of covering one’s mouth when coughing, followed by handwashing [1180/1726 (68%)], and wearing face masks [520/1726 (30%)] (Table 4).

### 3.5. Determinants of Level of Satisfaction with COVID-19 Preventive Measures in the First Stage of the Outbreak, Uganda

In multivariable analysis, females (AOR: 1.3, 95% CI: 1.1–1.6), health care workers (AOR: 1.2, 95% CI: 1.02–1.5), and those in the second wealth quintile (AOR: 1.4, 95% CI: 1.02–1.9) were very satisfied with the preventive measures (Table 5). Participants who reported violence or discrimination at home during the lockdown period (AOR: 0.25, 95% CI: 0.09–0.67) were less likely to be very satisfied with the COVID-19 the preventive measures.

## 4. Discussion

This study assessed adherence to and satisfaction with COVID-19 prevention measures in the early phase of the outbreak in Uganda. Only 29% adhered to all preventive measures of interest, although adherence to some measures was very high. Nearly all participants (96%) reported frequent handwashing with soap, but only 33% reported wearing a face mask when going out. It has been estimated that proper masks use with a coverage of 80% would halt the transmission of the virus [5]. However, like other countries in Africa, masking is not commonly done and was only introduced in response to the COVID-19 pandemic. Low usage of masks could also be a result of the initial inconsistency in information about the value of mask use by the general population to prevent COVID-19 transmission [6]. Additionally, there was information that the threat of COVID-19 posed to Africa and Uganda would be mild, given the tropical environment and the largely young population structure [7]. Furthermore, many Africans did not wear a mask because it was uncomfortable, or because they did not even think that it was necessary [8]. At the start of this study, Uganda had only 55 cases and 0 deaths due to COVID-19. The numbers increased to 63 cases by the end of the study, still with no fatalities. An exponential increase in the number of reported COVID-19 cases did not occur despite the low adherence to personal preventive measures. The reasons for this low COVID-19 morbi-mortality warrant further investigation. Although lockdown measures and cross border movement restrictions [9] may have contributed to downplaying the local COVID-19 outbreak, recent findings suggest that in most African countries, even where less strict lockdown measurements were implemented, the COVID-19 outbreak did not seem to cause a dramatic increase in mortality [10]. However, more sensitization regarding the importance of face masks use in containing the COVID-19 pandemic is clearly needed as well as subsidies and free masks for those who may not be able to afford them.

The low levels of adherence revealed in this study could imply that the compliance to the government preventive measures could have potentially further declined during the course of the outbreak, reflecting the need to upscale risk communication strategies in the COVID-19 response and future similar outbreaks. Additionally, enforcement of preventive measures, such as wearing masks, hand hygiene, and physical distancing, in the population, could stabilize the outbreak and halt the viral transmission. When the Arizona state in the USA enforced mask use and other preventive measures, cases reduced by 75% in about a month [11].

Living in Kampala City Centre was associated with high adherence to preventive measures. This is probably explained by the fact that the first cases of COVID-19 were reported in Kampala and that people in Kampala were more exposed to information about COVID-19 than elsewhere. Respondents who reported living in a household with other siblings were less likely to adhere to the preventive measures. This could be because some of the siblings are young people, thus have a low-risk perception of COVID-19 [12], and for them, physical distancing may be difficult. In addition, larger families may have more financial and space constraints.

Receiving COVID-19 related information from health workers was also associated with good preventive behavior. The country’s Ministry of Health, through its decentralized systems, used health workers to sensitize the public on COVID-19 through various fora, including community outreach. The population is more likely to trust information from health workers and any other trusted source [13].

Worry about one’s health was also associated with high adherence to preventive measures. This concurs with findings from a Canadian study, which described how concerns about health status may be associated with adherence to disease preventive measures [14]. Risk perception is indeed an important determinant of the adoption of health promotion and preventive measures. However, in Uganda, health promotion to prevent COVID-19 transmission has been a major challenge due to widespread misinformation and disinformation, which downplayed the risk of COVID-19 [12].

Satisfaction with preventive measures was associated with increased adherence. This is not surprising but also highlights the need to ensure that trust and satisfaction are maintained to sustain adherence to government interventions [15]. This, coupled with the perception of the effectiveness of COVID-19 preventive measures, should be integrated within the COVID-19 risk communication and community engagement, especially for the men who reported lower satisfaction and adherence levels compared to the women [15,16]. Men generally have more challenges, poorer health-seeking behaviors, and less contact with the healthcare system [17]. Of note, participants who experienced violence reported lower satisfaction, perhaps because the violence could have been related to enforcement of the preventive measures [18]. Punitive measures in ensuring adherence to COVID-19 measures is an emerging area of concern that has not been fully explored and requires more research.

## 5. Limitations of the Study

The study was conducted online, and this required access to smartphones and internet connectivity for participation in the survey. The study sample may not be demographically representative enough as it comprised mostly educated people with a certain social standing. Therefore, our findings could have overestimated the level of adherence and satisfaction.

## 6. Conclusions

Relatively low proportions of respondents adhered to all the recommended preventive measures, and adherence was especially low concerning the use of masks. The proportion of respondents who were very satisfied with preventive measures was also low. Behavior change programs need to be intensified to improve the level of adherence and satisfaction with preventative measures, especially the use of masks. Special messages and efforts should target men, large families, and people living outside Kampala city center and be popularized at the community level by health workers and community leaders.

## Figures and Tables

**Table 1 ijerph-17-08810-t001:** Socio-demographic characteristics of study participants.

Characteristic (*n* = 1726)	Survey Findings
**Age groups**	
<18 years, *n* (%)	13 (0.75%)
18–28 years, *n* (%)	445 (26%)
29–39 years, *n* (%)	706 (41%)
40–49 years, *n* (%)	347 (20%)
50+ years, *n* (%)	215 (13%)
**Sex**	
Female, *n* (%)	711 (41%)
Male, *n* (%)	1015 (59%)
**Nationality**	
Ugandan, *n* (%)	1679 (97%)
Foreigner, *n* (%)	47 (3%)
**Religion**	
Muslim, *n* (%)	97 (5.6%)
Catholic, *n* (%)	536 (31%)
Protestant, *n* (%)	631 (37%)
Pentecostal, *n* (%)	320 (19%)
Seventh Day Adventist & others, *n* (%)	102 (5.9%)
Non-religious, *n* (%)	40 (2.3%)
**Education level**	
University Postgraduate Degree (Masters & PhD), *n* (%)	797 (46%)
Tertiary (Certificate, diploma and degree), *n* (%)	863 (50%)
Secondary, *n* (%)	63 (3.7%)
Primary and No education, *n* (%)	3 (0.17%)
**Marital status**	
Cohabitation, *n* (%)	247 (14%)
Divorced, *n* (%)	30 (1.7%)
Legally married, *n* (%)	754 (44%)
Single, *n* (%)	676 (39%)
Widow/widower, *n* (%)	19 (1.1%)
**Place of residence**	
Rural area/village, *n* (%)	189 (11%)
Kampala suburb, *n* (%)	688 (40%)
Kampala city center, *n* (%)	186 (11%)
Other town/city suburb, *n* (%)	334 (19%)
Other town/city center, *n* (%)	329 (19%)
**Occupation**	
Jobless, *n* (%)	124 (7.2%)
Self-employed, *n* (%)	284 (17%)
Student, *n* (%)	209 (12%)
Work for a person, institution, or company, *n* (%)	731 (42%)
Work for the government, *n* (%)	378 (22%)
**Being a Health worker**	
No, *n* (%)	1108 (64%)
Yes, *n* (%)	618 (36%)
**Living alone**	
No, *n* (%)	1479 (86%)
Yes, *n* (%)	247 (14%)
**Wealth Index quintile**	
Lowest, *n* (%)	350 (20%)
Second, *n* (%)	351 (20%)
Middle, *n* (%)	343 (20%)
Fourth, *n* (%)	361 (21%)
Highest, *n* (%)	321 (19%)
**Underlying disease**	
Known underlying disease, *n* (%)	300 (17%)
No known underlying diseases, *n* (%)	1426 (83%)

*n*: number of participants in a particular response category.

**Table 2 ijerph-17-08810-t002:** Level of adherence to Coronavirus Disease 2019 (COVID-19) preventive measures.

Variables (*n* = 1726)	Response	Survey Findings
Observe physical distance	No, *n* (%)	171 (10)
	Yes, *n* (%)	1555 (90)
Wear face mask when going outside	No, *n* (%)	1160 (67)
	Yes, *n* (%)	566 (33)
Cover mouth when coughing or sneezing	No, *n* (%)	248 (14)
	Yes, *n* (%)	1478 (86)
Frequent handwashing	No, *n* (%)	64 (3.7)
	Yes, *n* (%)	1662 (96)
Stay home when feel flu-like symptoms	No, *n* (%)	293 (17)
	Yes, *n* (%)	1433 (83)
Avoiding touching face	Yes, *n* (%)	1344 (78)
	No, *n* (%)	382 (22)
Disinfecting phone	Yes, regularly, *n* (%)	416 (24)
	Yes, whenever I return home, *n* (%)	317 (18)
	No, *n* (%)	993 (58)
Disinfecting Laptop	Yes, regularly, *n* (%)	243 (14)
	Yes, before use, *n* (%)	201 (12)
	No, *n* (%)	1282 (74)
Disinfecting door locks	Yes, regularly, *n* (%)	331 (19)
	Yes, before use, *n* (%)	152 (8.8)
	No, *n* (%)	1243 (72)
Disinfecting TV remote	Yes, regularly, *n* (%)	208 (12)
	Yes, before use, *n* (%)	102 (5.9)
	No, *n* (%)	1416 (82)
Disinfecting bag handles	Yes, regularly, *n* (%)	220 (13)
	Yes, before use, *n* (%)	134 (7.8)
	No, *n* (%)	1372 (80)
Approximate number of times hands were washed or hand sanitizer used during the past day, median (IQR)		6 (5–10)
Overall score of adherence to preventive measures ^1^	Very poor adherence, *n* (%)	66 (3.8)
	Poor adherence, *n* (%)	274 (16)
	Moderate adherence, *n* (%)	891 (52)
	High adherence, *n* (%)	495 (29)

^1^ It is a composite variable generated from the major four selected COVID-19 preventive measures. *n*: number of respondents.

**Table 3 ijerph-17-08810-t003:** Determinants of adherence to COVID-19 preventive health measures in the first stage of the outbreak, Uganda.

Variables	Frequency (Percentage)				Unadjusted OR (95% CI)	Adjusted OR (95% CI)
	Very Poor Adherence	Poor Adherence	Moderate Adherence	High Adhere		
Sex						
Female	30 (4.2)	98 (14)	348 (49)	235 (33)	1.3 (1.1–1.6) ***	1.2 (0.95–1.4)
Male	36 (3.6)	176 (17)	543 (54)	260 (26)	Ref	Ref
Age group						
18–28 years	2 (15)	2 (15)	5 (38)	4 (31)	1.5 (0.5–4.8) ***	1.7 (0.51–5.6)
29–39 years	17 (3.8)	65 (15)	226 (51)	137 (31)	1.3 (0.4–4.1) ***	1.4 (0.42–4.6)
40–49 years	30 (4.3)	113 (16)	372 (53)	191 (27)	1.4 (0.5–4.4) ***	1.4 (0.42–4.7)
50+ years	14 (4.0)	51 (15)	185 (53)	97 (28)	1.5 (0.5–4.6) ***	1.4 (0.40–4.6)
<18 years	3 (1.4)	43 (20)	103 (48)	66 (31)	Ref	Ref
Residence						
Kampala suburbs	24 (3.5)	126 (18)	313 (46)	225 (33)	1.1 (0.82–1.5)	1.3 (0.93–1.9)
Kampala city centre	4 (2.2)	25 (13)	83 (45)	74 (40)	1.6 (1.1–2.4) *	1.7 (1.1–2.6) *
Other city/town suburbs	9 (2.7)	44 (13)	212 (63)	69 (21)	0.92 (0.66–1.3)	1.1 (0.75–1.6)
Other cities/towns centre	15 (4.6)	50 (15)	195 (59)	69 (21)	0.84 (0.59–1.2)	0.95 (0.67–1.4)
Rural area/village	14 (7.4)	29 (15)	88 (47)	58 (31)	Ref	Ref
COVID-19 information from health care workers						
Yes	26 (3.3)	109 (14)	403 (51)	253 (32)	1.4 (1.1–1.6) ***	1.2 (1.0–1.5) *
No	40 (4.3)	165 (18)	488 (52)	242 (26)	Ref	Ref
COVID-19 information from village leaders						
Yes	5 (2.9)	17 (9.8)	91 (52)	61 (35)	1.5 (1.1–2.0) **	1.4 (1.0–1.9) *
No	61 (3.9)	257 (17)	800 (52)	434 (28)	Ref	Ref
COVID-19 information from television						
Yes	48 (3.3)	232 (16)	753 (51)	433 (30)	1.3 (1.1–1.7) *	1.3 (1.0–1.7)
No	18 (6.9)	42 (16)	138 (53)	62 (24)	Ref	Ref
Religion						
Seventh Day Adventist and other	2 (2.0)	12 (12)	55 (54)	33 (32)	2.5 (1.2–4.9) *	1.9 (0.95–4.0)
Pentecostal	10 (3.1)	68 (21)	160 (50)	82 (26)	1.6 (0.84–2.9)	1.3 (0.69–2.5)
Protestant	24 (3.8)	98 (16)	316 (50)	193 (31)	2.0 (1.1–3.7) *	1.7 (0.93–3.2)
Catholic	22 (4.1)	73 (14)	293 (55)	148 (28)	1.9 (1.1–3.5) *	1.6 (0.84–3.0)
Muslim	6 (6.2)	12 (12)	47 (48)	32 (33)	2.2 (1.1–4.4) *	1.9 (0.92–3.9)
Non-religious	2 (5.0)	11 (28)	20 (50)	7 (18)	Ref	Ref
Marital status						
Living as a couple	35 (3.5)	173 (17)	525 (52)	268 (27)	0.83 (0.69–0.99) *	1.1 (0.88–1.3)
Not living as a couple	31 (4.3)	101 (14)	366 (50)	227 (31)	Ref	Ref
Being a health worker						
Yes	19 (3.1)	80 (13)	331 (54)	188 (30)	1.25 (1.0–1.5) *	1.1 (0.88–1.3)
No	47 (4.2)	194 (17)	560 (51)	307 (28)	Ref	Ref
Working conditions						
Worker from home	29 (4.4)	112 (17)	352 (53)	170 (26)	0.93 (0.74–1.2)	1.01 (0.79–1.3)
Worker in a closed indoor space alone	6 (3.1)	23 (12)	98 (51)	65 (34)	1.4 (1.0–1.9) *	1.4 (0.99–2.0)
Worker in a closed indoor space with several people	8 (2.7)	48 (16)	145 (48)	99 (33)	1.2 (0.94–1.6)	1.2 (0.86–1.6)
Worker in an open space	3 (2.5)	20 (17)	57 (48)	38 (32)	1.2 (0.81–1.8)	1.3 (0.85–1.9)
Not applicable (if jobless or student)	20 (4.4)	71 (16)	239 (53)	123 (27)	Ref	Ref
Worry about own health						
Moderately not worried	11 (4.1)	54 (16)	141 (53)	61 (23)	0.77 (0.59–1.0)	0.9 (0.68–1.2)
Worried	14 (4.2)	54 (16)	183 (54)	86 (26)	0.92 (0.72–1.2)	1.1 (0.83–1.4)
Moderately worried	5 (3.0)	18 (11)	88 (53)	55 (33)	1.3 (0.98–1.9)	1.4 (0.97–1.9)
Extremely worried	12 (4.0)	35 (12)	146 (49)	104 (35)	1.4 (1.1–1.8) *	1.5 (1.1–1.9) **
Not worried	24 (3.6)	113 (17)	333 (51)	189 (29)	Ref	Ref
Wealth Index quintile						
Second	13 (3.7)	49 (14)	202 (58)	87 (25)	1.1 (0.82–1.4)	1.03 (0.77–1.4)
Middle	9 (2.6)	59 (17)	174 (51)	101 (29)	1.2 (0.89–1.6)	1.1 (0.83–1.5)
Fourth	10 (2.8)	67 (19)	177 (49)	107 (30)	1.1 (0.87–1.5)	1.1 (0.79–1.5)
Highest	15 (4.7)	47 (15)	146 (45)	113 (35)	1.4 (1.1–1.9) *	1.3 (0.95–1.9)
Lowest	19 (5.4)	52 (15)	192 (55)	87 (25)	Ref	Ref
Living with siblings at home						
Yes	16 (3.6)	78 (17)	247 (55)	106 (24)	0.79 (0.64–0.97) *	0.75 (0.61–0.93) **
No	50 (3.9)	196 (15)	644 (50)	389 (30)	Ref	Ref
Level of satisfaction						
Dissatisfied	15 (5.0)	53 (18)	171 (57)	59 (20)	1.7 (1.3–2.4) ***	1.7 (1.2–2.3) **
Neutral	13 (4.6)	51 (18)	161 (57)	58 (20)	1.8 (1.3–2.4) ***	1.7 (1.3–2.4) ***
Satisfied	5 (1.6)	45 (15)	190 (61)	71 (23)	2.3 (1.7–3.2) ***	2.2 (1.6–3.0) ***
Very satisfied	6 (1.1)	47 (8.6)	230 (42)	262 (48)	6.0 (4.5–8.0) ***	5.6 (4.2–7.5) ***
Very dissatisfied	27 (9.3)	78 (27)	139 (48)	45 (16)	Ref	Ref

* <0.05; ** <0.01; *** <0.001 level of significance; OR: Odds ratio (Measure of association between outcome and exposure/independent variables); CI: Confidence interval; Ref: Reference category; Unadjusted OR: odds ratio that does not account for confounders; Adjusted OR: odds ratio that controls for other independent variables.

**Table 4 ijerph-17-08810-t004:** Level of satisfaction with COVID-19 preventive measures in the first stage of the outbreak, Uganda.

Variables	Survey Findings
Stay at home	
Extremely dissatisfied, *n* (%)	134 (7.8)
Dissatisfied, *n* (%)	133 (7.7)
Satisfied, *n* (%)	348 (20)
Moderately satisfied, *n* (%)	403 (23)
Extremely satisfied, *n* (%)	708 (41)
Frequent handwashing with soap	
Extremely dissatisfied, *n* (%)	42 (2.4)
Dissatisfied, *n* (%)	40 (2.3)
Satisfied, *n* (%)	133 (7.7)
Moderately satisfied, *n* (%)	331 (19)
Extremely satisfied, *n* (%)	1180 (68)
Physical distancing	
Extremely dissatisfied, *n* (%)	58 (3.4)
Dissatisfied, *n* (%)	76 (4.4)
Satisfied, *n* (%)	237 (14)
Moderately satisfied, *n* (%)	393 (23)
Extremely satisfied, *n* (%)	962 (56)
Wear a face mask	
Extremely dissatisfied, *n* (%)	240 (14)
Dissatisfied, *n* (%)	217 (13)
Satisfied, *n* (%)	420 (24)
Moderately satisfied, *n* (%)	329 (19)
Extremely satisfied, *n* (%)	520 (30)
Avoid spitting in the open space	
Extremely dissatisfied, *n* (%)	54 (3.1)
Dissatisfied, *n* (%)	28 (1.6)
Satisfied, *n* (%)	84 (4.9)
Moderately satisfied, *n* (%)	156 (9.0)
Extremely satisfied, *n* (%)	1404 (81)
Cover mouth or nose with tissue paper or fabric when coughing/ sneezing	
Extremely dissatisfied, *n* (%)	59 (3.4)
Dissatisfied, *n* (%)	39 (2.3)
Satisfied, *n* (%)	138 (8.0)
Moderately satisfied, *n* (%)	239 (14)
Extremely satisfied, *n* (%)	1251 (73)
Avoid meetings or gatherings of more than 5 people	
Extremely dissatisfied, *n* (%)	64 (3.7)
Dissatisfied, *n* (%)	51 (3.0)
Satisfied, *n* (%)	163 (9.4)
Moderately satisfied, *n* (%)	292 (17)
Extremely satisfied, *n* (%)	1156 (67)
Overall satisfaction with preventive measures ^1^	
Very dissatisfied, *n* (%)	289 (17)
Dissatisfied, *n* (%)	298 (17)
Neutral, *n* (%)	283 (16)
Satisfied, *n* (%)	311 (18)
Very satisfied, *n* (%)	545 (32)

^1^ It is a composite variable generated from the satisfaction level of each of the four major COVID-19 preventive measures. *n*: number of respondents.

**Table 5 ijerph-17-08810-t005:** Determinants of level of satisfaction with COVID-19 preventive measures in the first stage of the outbreak, Uganda.

Variables	Frequency (Percentage)					Unadjusted OR (95% CI)	Adjusted OR (95% CI)
	Very Dissatisfied	Dissatisfied	Neutral	Satisfied	Very Satisfied		
Sex							
Female	100 (14)	108 (15)	102 (14)	137 (19)	264 (37)	4.7 (1.3–1.8) ***	1.3 (1.1–1.6) **
Male	189 (19)	190 (19)	181 (18)	174 (17)	281 (28)	Ref	Ref
Age Group							
18–28 years	82 (18)	80 (18)	60 (18)	79 (18)	144 (32)	0.97 (0.36–2.6)	0.85 (0.31–2.3)
29–39 years	128 (18)	122 (18)	119 (17)	117 (17)	220 (31)	0.94 (0.35–2.5)	0.98 (0.36–2.7)
40–49 years	49 (14)	61 (18)	72 (21)	61 (18)	104 (30)	1.0 (0.37–2.7)	1.1 (0.39–3.0)
50+ years	27 (13)	34 (16)	30 (14)	51 (24)	73 (34)	1.3 (0.46–3.4)	1.4 (0.48–3.8)
<18 years	3 (23)	1 (8)	2 (15)	3 (23)	4 (31)	Ref	Ref
Wealth Index quintile							
Second	55 (16)	57 (16)	66 (19)	59 (17)	114 (32)	1.3 (0.99–1.7)	1.4 (1.02–1.9) *
Middle	48 (14)	64 (19)	53 (15)	68 (20)	110 (32)	1.3 (1.01–1.7)	1.4 (0.95–2.1)
Fourth	62 (17)	56 (16)	57 (16)	72 (20)	114 (32)	1.3 (0.98–1.7) **	1.3 (0.85–2.0)
Highest	46 (14)	62 (19)	50 (16)	59 (18)	104 (32)	1.3 (0.99–1.7) *	1.3 (0.79–2.0)
Lowest	78 (22)	59 (17)	57 (16)	53 (15)	103 (29)	Ref	Ref
Being a health care worker							
Yes	87 (14)	102 (17)	104 (17)	114 (18)	211 (34)	1.2 (1.0–1.5) *	1.2 (1.0–1.5) *
No	202 (18)	196 (18)	179 (16)	197 (18)	334 (30)	Ref	Ref
Working conditions							
Worker from home	124 (19)	116 (18)	120 (18)	119 (18)	184 (28)	0.86 (0.70–1.07)	1.0 (0.74–1.4)
Worker in a closed indoor space alone	25 (13)	35 (18)	27 (14)	52 (27)	53 (28)	1.1 (0.79–1.4)	1.3 (0.88–1.9)
Worker in a closed indoor space with several people	34 (11)	48 (16)	50 (17)	54 (18)	114 (38)	1.4 (1.0–1.8) *	1.6 (1.1–2.3)
Worker in an open space	22 (19)	23 (19)	17 (14)	15 (13)	41 (35)	0.96 (0.66–1.4)	1.3 (0.82–1.9)
Not applicable (if jobless or student)	84 (19)	76 (17)	69 (15)	71 (16)	153 (34)	Ref	Ref
Worry about loved ones’ health							
Moderately not worried	60 (22)	48 (18)	44 (16)	58 (22)	59 (22)	0.54 (0.42–0.71) ***	0.59 (0.44–0.78) ***
Worried	72 (20)	70 (19)	68 (19)	66 (18)	91 (25)	0.59 (0.46–0.75) ***	0.63 (0.48–0.82) **
Moderately worried	36 (16)	48 (21)	35 (15)	43 (19)	67 (29)	0.71 (0.54–0.94) *	0.77 (0.56–1.1)
Extremely worried	48 (14)	60 (17)	47 (14)	64 (19)	126 (37)	0.93 (0.73–1.2)	0.85 (0.63–1.1)
Not worried	73 (14)	72 (14)	89 (17)	80 (16)	202 (39)	Ref	Ref
Worry about own health							
Moderately not worried	47 (18)	52 (19)	48 (18)	57 (21)	63 (24)	0.69 (0.54–0.89) **	0.86 (0.65–1.1)
Worried	69 (24)	65 (19)	55 (16)	66 (20)	82 (24)	0.65 (0.51–0.82) ***	0.76 (0.60–1.0)
Moderately worried	19 (12)	36 (22)	28 (17)	37 (22)	46 (28)	0.86 (0.64–1.2)	0.97 (0.70–1.4)
Extremely worried	48 (16)	47 (16)	46 (15)	43 (14)	113 (38)	1.0 (0.79–1.2)	1.1 (0.80–1.5)
Not worried	106 (16)	98 (15)	106 (16)	108 (16)	241 (37)	Ref	Ref
Ever Suffered violence							
No	253 (16)	274 (17)	263 (17)	284 (18)	515 (31)	1.6 (1.2–2.2) **	1.2 (0.68–2.3)
Yes	36 (26)	24 (18)	20 (15)	27 (20)	30 (22)	Ref	Ref
Suffer violence and discrimination perpetrated by family members							
Yes	13 (45)	3 (10)	7 (24)	4 (14)	2 (6.9)	0.27 (0.14–0.54) ***	0.25 (0.09–0.67)**
No	276 (16)	295 (17)	276 (16)	307 (18)	543 (32)	Ref	Ref

* <0.05; ** <0.01; *** <0.001 level of significance; OR: Odds ratio (Measure of association between outcome and exposure/ independent variables); CI: Confidence interval; Ref: Reference category; Unadjusted OR: odds ratio that does not account for confounders; Adjusted OR: odds ratio that controls for other independent variables.

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
