# Peer review of "Level and Determinants of Adherence to COVID-19 Preventive Measures in the First Stage of the Outbreak in Uganda"

_ijerph, 2020, doi:10.3390/ijerph17238810_

Round 1
Reviewer 1 Report
Manuscript submitted by Amodan et al., entitled as “Level and determinants of adherence to COVID-19 preventive measures in the first stage of the outbreak in Uganda” details effectiveness of government measures to tackle COVID-19 and their perception among residents of Uganda. While the study provides a good understanding of people’s perception, but it lacks in understanding what more or different could have been done to make situation better. Since COVID-19 was unprecedented, therefore studies of this nature are very helpful in crafting future action if such a situation may arise again. Overall, manuscript is of good interest, prepared well. It should address following points to make it more acceptable.
- Authors should provide a detail of government issued guidelines implemented to prevent or minimize viral transmission. How those guidelines were suited to country specific needs of Uganda. In the discussion section authors should provide their perspective, based on the findings of this study, how and what measures could be taken to make these guidelines more effective and appealing.
- Authors need to provide better details of statistical methods used. Each table, in the footnote, should describe those methods. Also, state meaning of unadjusted and adjusted values and how they were achieved. It will be helpful to readers without statistical background.
- Authors should provide more detail on various variables, and what does those indicate. E.g. wealth index in Table 5, how was it assessed? A US$ value can be provided to make it better understood by larger section of readers.
- It is not clear in what language the survey was administered. Also, a link or proper reference to ICP questionnaire and details of modifications with rationale will be helpful.
- How responders were targeted? What was the source of Email list and what social media platform were used and this survey promoted? Those details will be helpful for future researchers.
- Response from 1726 persons was recorded. How does it represent the country overall? What is the total population and accordingly what is the distribution of these responders? Are these responses really represent perception of an entire country, if not (as authors have pointed out in their study limitation) then how perception of government policy could be measured. Authors can cite some past studies to get an idea of “power analysis” required to get an idea of impact of a government policy.
Author Response
Dear Reviewer 1,
We are grateful for the opportunity to revise our our manuscript. Your comments were very helpful in improving the paper in a substantial manner.
Please find attached details of our response.
Thank you.

Reviewer 2 Report
Dear authors,
Thank you for your contribution to understand Covid 19 mitigating measures better.
I find your manuscript of interest, generally well designed and presented. However, I have some suggestions for further improvement of the study and in particular the presentation of it.
Introduction: Nothing to add.
Results: I suggest that you use a figure to illustrate the results, or change format of the Table to a
Measure Yes (n, %) No (n,%). As it is now, it is very hard go understand what answers that belongs to each row.
Table 3 has the heading “Determinants of adherence”. I guess this is the results of the multiple regression
calculation you mention above this Table? If so, I suggest you to add what factors analysed that were significant, and the impact of each value on the outcome variable. Otherwise, please use a traditional multifactor regression for this analys.
You have analyzed the Level of satisfaction with the COVID-19 preventive health measures. I wonder why this is interesting? What is satisfaction? You can be very satisfied with regulations and preventive measures if you find them relevant, or if you feel safe from them, but you can also be very satisfied if you are not affected in your daily life- even if the measures are not at all effective. To me, this is a strange question and I think the intepretatiosn of the results may be difficult, if not using a mixed methods analysis based on both these results but also interviews to clarify the answers.
Discussion: It would add value if you could comment your results in the light of the number of infected or deceased in Covid 19 in Uganda at the time for this survey.
Limitations: You need to comment on the sample. As I understand, this is a convenient sample? Is the demographic data in the study population in line with a representative sample from the whole population?
Author Response
Dear Reviewer 2,
We are grateful for the opportunity to revise our our manuscript. Your comments were very helpful in improving the paper in a substantial manner.
Please find attached details of our response.
Thank you

Round 2
Reviewer 2 Report
Thank you for your efforts to further imporve the manuscirpt. I still think the Tables are very hard to read, but I leave this for the editorial services to check on. Snowballing is a sampling method mainly used qualitative researh, I would prefer to call this sampling a convinient sample and suggest that you change this. Otherwise, I´m satisfied with the manuscirpt.
Author Response
Response to Reviewer 2 Second round Comments
Point 1: "Thank you for your efforts to further imporve the manuscirpt. I still
think the Tables are very hard to read, but I leave this for the
editorial services to check on. Snowballing is a sampling method mainly
used qualitative researh, I would prefer to call this sampling a
convinient sample and suggest that you change this. Otherwise, I´m
satisfied with the manuscirpt."
Response 1: We agree with you that snowballing is a method normally used in qualitative methods. But also, we did not choose a convenient sample or sampling technique to get our participants. We targeted to share the link with all those with smart phones and internet connectivity in the whole country. We therefore asked everyone who got a link to also share it with their networks. Of course, we did get everyone participate but the link was spread widely in the whole country. See minor revisions in line….79-80